# An Edge Computing and Ambient Data Capture System for Clinical and Home Environments

**DOI:** 10.3390/s22072511

**Published:** 2022-03-25

**Authors:** Pradyumna Byappanahalli Suresha, Chaitra Hegde, Zifan Jiang, Gari D. Clifford

**Affiliations:** 1School of Electrical and Computer Engineering, Georgia Institute of Technology, Atlanta, GA 30332, USA; pradyumna@gatech.edu (P.B.S.); chegde@gatech.edu (C.H.); 2Department of Biomedical Informatics, Emory University, Atlanta, GA 30322, USA; 3Department of Biomedical Engineering, Georgia Institute of Technology, Emory University, Atlanta, GA 30332, USA; zifanjiang@gatech.edu

**Keywords:** Raspberry Pi, edge computing, ambient health monitoring, privacy-preserving, bluetooth, geolocation tracking, patient alarm, illuminance

## Abstract

The non-contact patient monitoring paradigm moves patient care into their homes and enables long-term patient studies. The challenge, however, is to make the system non-intrusive, privacy-preserving, and low-cost. To this end, we describe an open-source edge computing and ambient data capture system, developed using low-cost and readily available hardware. We describe five applications of our ambient data capture system. Namely: (1) Estimating occupancy and human activity phenotyping; (2) Medical equipment alarm classification; (3) Geolocation of humans in a built environment; (4) Ambient light logging; and (5) Ambient temperature and humidity logging. We obtained an accuracy of 94% for estimating occupancy from video. We stress-tested the alarm note classification in the absence and presence of speech and obtained micro averaged F1 scores of 0.98 and 0.93, respectively. The geolocation tracking provided a room-level accuracy of 98.7%. The root mean square error in the temperature sensor validation task was $0.3{\;}^{{}^{\circ}}$C and for the humidity sensor, it was 1% Relative Humidity. The low-cost edge computing system presented here demonstrated the ability to capture and analyze a wide range of activities in a privacy-preserving manner in clinical and home environments and is able to provide key insights into the healthcare practices and patient behaviors.

## 1. Introduction

Over the years, sensor technologies have played a critical role in patient monitoring in clinical and home environments. Despite this, much of the captured data is poorly integrated for research and retrospective analysis. Moreover, certain key events occurring in these settings remain undocumented with any level of spatiotemporal precision.

In clinical environments such as operating rooms (ORs) and intensive care units (ICUs), key events during patient monitoring include: (1) Patient movements while lying in bed and in mobility within the room [1,2]; (2) Bedside monitor alarm triggers and noise pollution [2,3,4,5,6]; (3) Presence, absence and movement of clinical personnel in the patient’s vicinity [7,8,9]; and (4) Variations in the ambient light, temperature, and humidity [2,6,10]. In home environments, key events that are generally untracked but are beneficial for patient monitoring include: (1) Patient bodily movement during sleep [11,12]; (2) Patient movement around their residence [13]; (3) Doorbell triggers, smoke-detector triggers, microwave beeps, and phone rings [14]; and (4) Changes in the ambient light, temperature, and humidity [15].

Recently, non-contact sensors or nearables [15,16,17,18,19,20] such as microphones, video cameras, light-intensity sensors, temperature and humidity sensors, are becoming more popular for hassle-free patient monitoring. They not only collect valuable patient behavior data but also pick up key information about the patient’s ambient environment while not interfering with the patient’s day-to-day activities.

The objective of this study was to develop a non-contact data capture and archival system to capture patient behavior and ambient environment information. While obtaining patient behavior and ambient information is crucial in understanding the effects of healthcare practices on patient health, maintaining patient privacy is as important if not more. For this, we utilized the edge computing paradigm. In *edge computing*, algorithms are decentralized and moved closer to the point of data capture to reduce latency and bandwidth requirements. This paradigm can be defined as computing outside the cloud, happening at the edge of the network, specifically in applications where real-time data processing is required. In our work, we utilized a Raspberry Pi (RPi) as a hub for data collection and edge computing. We extracted patient privacy-preserving features from the captured data on the RPi (at the edge) before discarding the raw underlying signal and transferred the computed features to a Health Insurance Portability and Accountability Act (HIPAA) compliant storage. The retrospective patient state analysis utilizing these captured features occurred on a central server away from the RPi.

Non-contact monitoring of patients is becoming more prevalent, especially in elderly patients [21,22,23] and neurodevelopmental populations (such as Autism Spectrum Disorder) [24] as these systems cause no burden on the patients in terms of wearing and operating the device (in contrast to a wearable such as the smart watch). Further, non-contact monitoring allows for monitoring the patient’s global movements in contrast to wearables, thus providing additional information about patient behaviors. The advent of the COVID-19 pandemic further increased the need for such systems [19] as they allow passive patient monitoring from a distance. However, there are multiple challenges in building such a system. First challenge lies in integrating different sensors to capture multiple data modalities under a single clock. Second, this system should asynchronously transfer the data to a HIPAA-compliant database. Lastly, the system should maintain patient privacy while capturing the various data modalities. We overcame these challenges by developing a novel software system that ran on an RPi. Using this system we integrated the following five sensors: (1) Passive Infrared (PIR) sensor (2) RPi-Infrared (IR) camera (3) Universal Serial Bus (USB) Microphone (4) TCS34725 color sensor and (5) DHT22 temperature-humidity sensor. We integrated a Coral USB Tensor Processing Unit (TPU) accelerator to perform compute intensive feature extraction task such as human pose estimation from video. Further, we utilized the onboard Bluetooth receiver to geolocate humans using Bluetooth beacons. The main novelty of our work lied in *capturing privacy-preserved features simultaneously from multiple sensors to perform human movement detection, registering auditory cues, human geolocation, and ambient environment monitoring*. It was a specific design consideration that all hardware could be easily purchased at a low cost. This effectively helped scale the system and enabled us to capture data in large clinical environments. Further, the system was intended to be deployed rapidly without the need for expert fabrication of hardware.

We have described five applications of the data captured by our system. Namely: (1) Estimating occupancy and human activity phenotyping, (2) Medical equipment alarm classification, (3) Geolocation of humans in a built environment, (4) Ambient light logging, and (5) Ambient temperature and humidity logging. We provide the details of these experiments in Section 3.3. While estimating occupancy, activity phenotyping and alarm classification tasks aid us in non-contact patient monitoring, the management of indoor air temperature, humidity, and light intensity are vital for maintaining patients’ comfort in hospitals [25]. In [26], researchers recommended creating separate thermal zones according to patients’ preferences for improved recovery. Further, a relative humidity (RH) greater than 45–50% assists fungal growth in built environments and can affect patient recovery in hospitals [27]. With respect to the lighting conditions, patients prefer a mixture of warmer and cooler luminaire throughout the day and favor distributed lighting over the traditional over-bed configuration [28]. Thus, temperature, humidity, and light intensity monitoring are essential components in our ambient data capture system. These sub-parts assist clinicians and caretakers make critical decisions about the patient’s environmental conditions.

## 2. Related Work

### 2.1. Wearables and Body Area Networks

Since the turn of the 21st century, wearables (on-body sensors) have been at the forefront of non-traditional health monitoring systems [29,30,31]. These sensors collect high-resolution physiological signal data such as the electrocardiogram (ECG) and galvanic skin response. Further, multiple wearables can be used in a network for remote patient monitoring. Body area networks (BANs) are one such system where multiple wearables continuously monitor human physiology and track the patient’s health status [32]. BANs utilize wireless technologies including the ultra-wideband [33], Bluetooth [34], and Zigbee [34,35] for this purpose. Although BANs capture high-resolution information regarding human physiology, they suffer from the following drawbacks:BAN sensors perform localized measurements. For example, a wrist-worn accelerometer measures the acceleration of hand/wrist (local) and does not reliably measure overall body movement (global). Although a network of accelerometers will alleviate this problem, it comes at the cost of causing inconvenience to the patient as they have to wear multiple sensors on their body over long periods.Data from BAN sensors are often corrupted by missing data due to motion artifacts [36] (the patient-induced noise in physiological signals by voluntary or involuntary bodily movements) and low compliance by the patients [37]. Human bodily movement causes motion artifacts in the physiological signals (say ECG) being captured by the wearable and thus leads to data degradation. Further, the wearer (a human) has to comply with a data acquisition schedule and follow the instructions diligently to generate *good quality* data.

### 2.2. Non-Contact Health Monitoring Systems

Non-contact health monitoring systems, on the other hand, capture global signals (e.g., overall body movements via video camera [16]) and are less dependent on patient compliance for data capture. A popular way of performing non-contact monitoring of patients is to use the Doppler radar technology [38,39]. The Doppler radar is a specialized radar system that can measure target displacement remotely by using the Doppler effect. It has been used for gait-assessment of older adults [40], capturing human respiration signal [41] and human vital sign measurement [42]. While it does an excellent job of detecting body movements and measuring the vital signs of a patient, it does not capture auditory cues and other ambient environmental signatures. Thus, it does not suit our needs.

Numerous works have proposed the usage of some of our system’s individual components for patient monitoring. Extensive research has been performed to study the effect of noise pollution on patient and staff health, the performance of staff, and patient safety in clinical environments [43,44,45,46,47]. However, very few works describe methods to capture privacy-preserving ambient sound in clinical and home environments. In particular, Guerlain et al. [48] presented a methodology for archiving multi-channel audio and video recordings of OR during surgeries to facilitate prospective studies of operative performance. In contrast, our system avoided capturing raw audio signals. We developed a method to compute useful audio features from the captured raw audio on the RPi, archive the feature vectors, and discard the raw audio signals.

To geolocate clinical personnel at fine-resolution, Azevedo-Coste et al. [49] proposed using multiple cameras installed in ORs along with a wireless network of inertial sensors. On the other hand, recently, there has been a surge in the development of radio frequency (RF)-based non-contact human movement detectors and geolocators [50,51,52]. However, to keep the system low-cost, we avoided using these components in our edge computing system. Instead, our method utilized PIR sensors and movement and pose signals from RPi-IR cameras to capture patient movements and avoided retaining the raw video signals. Further, we proposed performing geolocation in different environments (clinical and home) using Bluetooth technology.

The inclusion of methods to capture ambient environmental data was based on the evidence presented in the following studies. Aschoff et al. [53] provided a comprehensive explanation about the effect of ambient temperature, humidity, and light intensity on the human circadian rhythm and sleep. Further, there is a growing evidence [54,55,56,57,58,59] about the effects of ambient environment on patient health. While few systems [28,60] provided methodology for capturing ambient environmental measures, we did not find works that captured concurrent patient motion, auditory noise and environmental signals.

### 2.3. Applications of Non-Contact Monitoring Systems

Numerous existing studies describe applications similar to the ones described in our work, albeit in isolation. Specifically, human occupancy estimation has been performed using the thermal imager [61,62] and the RGB camera [63]. These methods estimated human occupancy when there were three to five people in the room. The human activity recognition task has been performed using the RGB camera [64,65,66] and the thermal imager [67] as well. The number of people in these studies varied from three to 17 and the number of activities recognized varied from three to six. The geolocation of humans has been performed using Bluetooth beacons and inertial measurement unit (IMU) sensors. Sato et al. [68] performed geolocation on six participants using the range-only extended Kalman filter simultaneous localization and mapping technique. Martín et al. [69] utilized the IMU sensor along with the Bluetooth beacon to obtain a room level (six-rooms) human localization while geolocating four participants. An extensive literature survey reviled no studies that discussed classification of the medical equipment alarm notes in clean and noisy (background speech) environments. However, there were multiple works that analyzed generic ambient environmental sounds including in-home sounds, emergency sirens and everyday sounds. We have described three specific studies that were related to our work on alarm note classification. Jain et al. [70] developed two prototype systems for in-home sound analysis and deployed each system in four different homes to recognize 23 in-home sounds and three outdoor sounds. Cantarini et al. [71] applied the harmonic percussive source separation technique to classify emergency siren sounds from road noise sounds. Wyatt et al. [72] deployed a BERT-based environmental sound classification model on an RPi Zero to identify six different everyday sounds (Knock, Laugh, Keyboard Typing, Cough, Keys Jangling and Snap). All three studies mentioned above described standalone audio capture and recognition systems for varied environmental settings and did not describe a multimodal data acquisition system. There are few works in the literature that perform ambient environment data capture. Rashed et al. [60] described a medical platform for remote healthcare and assisted living. They utilized a DHT11 temperature and humidity sensor and a TSL235R light to frequency converter to keep track of the patient’s environmental conditions.

## 3. Materials and Methods

### 3.1. System Architecture

The system architecture described here is a low-cost, high-compliance design. At its core is the Python script that interfaces with the sensors via the Raspberry Pi. A picture illustrating the hardware components of the system is shown in Figure 1. The bill-of-materials, along with the total cost for the hardware components and the system dimensions, are provided in our open-sourced Github repository [73]. We now describe the system’s individual hardware components along with the associated software.

#### 3.1.1. Raspberry Pi

The RPi is a $35 computer that is about the size of a deck of cards. It functioned as the central hub in our data collection pipeline. In our work, we used the RPi 4 model B (Figure 1), which was released in June 2019 and was the most recent model during the software development stage of the project. The Debian-based operating system (‘Raspbian-Buster’) that is optimized for the RPi was installed for developers and users to interact with the hardware. Among the onboard peripherals on the RPi were two USB 2.0 ports and two USB 3.0 ports, a 40-pin General Purpose Input-Output (GPIO) header, and a USB-C port to supply power to the RPi. The RPi was powered using a 5 V 3 A power adapter. Alternatively, in low-resource settings, one can consider powering the Raspberry Pi using alternative means, for instance, a car battery [19]. In that case, a step-down transformer is required to convert a 12 V power supply to 5 V. However, in all our experiments, we used an external power supply that was available in the built environment.

#### 3.1.2. PIR Sensor Based Human Movement Detection

We used a PIR sensor (Figure 1) for coarse human movement detection. The PIR sensor consists of a pair of IR sensitive slots housed in a hermetically sealed metal casing. A Fresnel lens acts as the outermost cover, which increases the range and sensitivity of the sensor. When the sensor is idle, both the IR slots receive the same amount of IR radiation. Whereas, when an IR emanating object moves past the field of view of the first IR slot, this slot detects an increased IR radiation, and thus, a differential signal C between the two slots is generated. A differential signal C${}^{\prime }$, which is completely out of phase with respect to C, is generated when the object moves past the other IR-sensitive slot. These differentials are then processed to form the output signal. Our system was designed to capture data at a sampling frequency of 1Hz. The data itself was a binary spike train taking the value 1 when a movement was detected and 0 otherwise. In [12] we presented the method for capturing movement data using a PIR sensor during sleep from a single person. Further, we used the captured movement data to build a binary classifier for obstructive sleep apnea classification and obtained a classification accuracy equal to 91% [12]. In the current work, we extended this method and generalized PIR sensor-based human movement detection to function in different clinical and home environments. Based on the positioning and orientation of the sensor, one could capture different information. For instance, the timestamps when the patient’s knee was operated on could be obtained by placing the sensor to monitor a patient’s knee during their knee replacement surgery.

#### 3.1.3. IR Camera-Based Human Movement Detection

It is possible to use a video feed [16] from an RPi-IR camera in place of the PIR sensor to perform human movement detection. This method allowed us to capture the human movement signal with more than two quantization levels and obtain a finer signal than the binary signal captured using the PIR sensor. Besides capturing the occurrence of movements, the RPi-IR camera-based analysis allowed us to compute the intensity and direction of these movements. We used the **N**o **I**nfra**r**ed (NoIR) version [74] of the RPi-IR camera (Figure 1). In contrast to the regular RPi-camera, the NoIR RPi-IR camera did not employ infrared filters and gave us the ability to see in the dark with infrared lighting.

We now describe the method to extract human movement signal from the RPi-IR camera feed and describe its utility. Without loss of generality let us assume we capture the video at 1 Hz (i.e., 1 frame per second). Let the video frame at time *t* (in seconds) with a pixel resolution equal to M × N be denoted as $F_{t}$, and the previous video frame be $F_{t-1}$. The frame difference between the two frames at time *t* ($D_{t}$) is defined as $D_{t}=F_{t}-F_{t-1}$. The difference-frame $D_{t}$ has the same pixel resolution as $F_{t}$ and $F_{t-1}$, i.e., M × N. For a given video *V*, the corresponding difference frame-stack is given by the set $D={\left\{D_{t}\right\}}_{t\in [2,T]}$ where *T* is the total number of frames in *V* and $D_{t}$ is the difference frame at *t* seconds. We extract four different signals from *D* namely: (1) Global Difference Sum (GDS); (2) Global δ-Pixel Count (GDPC); (3) Local Difference Sum (LDS); and (4) Local δ-Pixel Count (LDPC). The global signals (GDS and GDPC) for a given difference frame $D_{t}$ are computed as follows:(1)$$\begin{array}{cc} GDS[t] & =\sum_{i\in [1,M]}\sum_{j\in [1,N]}\left|D_{t}\left[i,j\right]\right| \end{array}$$(2)$$\begin{array}{cc} GDPC[t] & =\#(\{d\in D_{t}\;|\;|d|>\delta \}) \end{array}$$
where *d* denotes an individual pixel in the difference frame $D_{t}$, |.| denotes the absolute value and #(.) denotes the set cardinality. GDS[t] is the sum total of the absolute values of the pixels in the difference frame at time *t* seconds and GDPC[t] is the total number of pixels in the difference frame at time *t* seconds that have an absolute value greater than δ. If δ=0, GDPC[t] denotes the total number of non-zero pixels in the difference frame $D_{t}$. The local signals (LDS and LDPC) for a given difference frame $D_{t}$ are computed likewise to the global signals but are calculated on smaller blocks in the difference frame. For this, we divide a difference frame of pixel resolution M × N into *K* parts of equal size along the M-axis and *L* parts of equal size along the N-axis. Each of the *K* parts contains $m=\frac{M}{K}$ columns and each of the *L* parts contains $n=\frac{N}{L}$ rows. This division along the rows and columns of a difference frame $D_{t}$ creates nBlocks=K×L local-blocks of size m×n. Let the $s^{th}$ local-block be denoted as $DL_{t,s}$. Then the local signals (LDS and LDPC) at time *t* for the $s^{th}$ local-block are given as:(3)$$\begin{array}{cc} LDS[t,s] & =\sum_{i\in [1,m]}\sum_{j\in [1,n]}\left|DL_{t,s}\left[i,j\right]\right| \end{array}$$(4)$$\begin{array}{cc} LDPC[t,s] & =\#(\{d\in DL_{t,s}\;|\;|d|>\delta \}) \end{array}$$
where *d* denotes an individual pixel in the difference frame local-block $DL_{t,s}$, |.| denotes the absolute value and #(.) denotes the set cardinality. The LDS[t,s] is the sum total of the absolute values of the pixels in the $s^{th}$ local-block of the difference frame at time *t* seconds and the LDPC[t,s] is the total number of pixels in the $s^{th}$ local-block of the difference frame at time *t* seconds that have an absolute value greater than δ. Similar to GDPC, if δ=0, LDPC[t,s] denotes the total number of non-zero pixels in the $s^{th}$ local-block of the difference frame $D_{t}$. The different parameters that needed to be set were δ,K,andL. The default values we set in our work were δ=0, K=5, L=4 and the videos we experimented on had a pixel resolution equal to 320×240. Thus, in our work, we had m=64, n=60, and nBlocks=20.

Together, the host of time-series data described above gave us information about the temporal and spatial variations in the video *V*. We needed certain assumptions so that these signals gave us information about human movement. The assumptions were as follows (1) We had one person in the entire video. (2) The background was static, and the only moving object in the video was the person or an object attached to the person. (3) The video was not corrupted or affected by noise.

While the PIR sensor provided binary movement signals (movement occurred vs. no movement), the movement signals from the IR cameras were finer and had a higher spatial resolution. These two solutions were useful in different scenarios. For instance, while the PIR sensor could be used to detect the presence or absence of a human in a room, the movement signal from the IR camera could be used to perform a privacy-preserving analysis of a patient’s sleep. Thus, based on the requirement and budget, the users of our system could use one or both sensors. For instance, if the application is human movement detection and cost is a concern, we recommend the users to use a PIR sensor.

#### 3.1.4. Human Pose and Activity Phenotyping

Technologies such as PoseNet [75,76], OpenPose [77] or DeepLabCut [78] could be used to obtain canonical representations of the human form, and when tracked over time, one could obtain information concerning pose and types of activities. In our implementation, we used a *Coral USB TPU Accelerator* (Figure 1) and Tensorflow Lite to render these abstractions in real time.

This data representation is known as keypoint representation and comprises the x-y coordinate positions of various interest points on the body, including the knees, elbows, and eyes. It has the advantage of preserving the privacy of individuals while also reducing the dimensionality of the data. One can use this technology to understand behavior of the neurodiverse populations, both at a group level and at an individual level, to capture social interaction metrics and predict certain behaviors of interest ahead of time.

In its implementation, we used a Google Coral Accelerator USB device in addition to the RPi and the RPi camera module V2 (8 megapixel). The RPi 4 was most suitable for this work compared to the older versions of the RPi due to the presence of USB-A 3.0 ports. These ports ensured fast communication between the Coral device and RPi. The absence of USB-A 3.0 ports in previous versions of the RPi significantly increased the communication time between the RPi and Coral, thus increasing the total run time of the algorithm implemented on it. We implemented the Tensorflow Lite model of the PoseNet algorithm on the Google Coral Accelerator that was connected to the USB-A 3.0 port of the RPi 4. The camera attached to the RPi collected videos, which were converted to the keypoint representations in real-time by PoseNet. These keypoint representations were stored on a secure server for further processing, and the raw video was discarded, thus achieving patient privacy. An example is given in Figure 2.

#### 3.1.5. Privacy Preserving Audio Data Capture

The proposed audio data capturing system consisted of a USB microphone connected to the RPi. In this work, we tested the *Fifine Conference USB Microphone* (Figure 1). In order to record the audio signal, we used the *python-sounddevice* package available on GitHub [79,80] under the MIT License. Specifically, we modified the script *rec_unlimited.py* [79] to be able to continuously record audio data and perform audio feature computation at regular intervals. To extract features from these audio snippets, we used a 30 millisecond Hanning window with a 50% overlap. Feature computation was done using the *librosa* package available on Github [81] under the Internet Systems Consortium (ISC) license. We utilized the spectral representation method *stft* in the Core IO and Digital Signal Processing toolbox and the spectral features method *mfcc* in the feature extraction toolbox in order to compute short-time Fourier transform (STFT) and Mel-frequency cepstral coefficients (MFCC), respectively. Further, we used the filter bank construction method *mel* in the Filters toolbox to create a filter bank with 10 frequency bins. For each 30 millisecond window, we then computed the signal energy in different frequency bins by performing the following matrix multiplication:(5)E=MS
where *S* was the STFT coefficient vector for the current window, *M* was the mel filter mask matrix with each row corresponding to a different mask, and *E* was the signal energy in different frequency bins corresponding to the mel filter masks. Note that the default shape of mel filter masks was a triangle with the mask values summing to one. We further included scripts for computing sample entropy of the windowed audio snippets using the *mse.c* script, which was available on Physionet [82] under the GNU general public license. Further, we developed scripts to archive the computed audio features to a secure cloud storage and discard the underlying audio snippets. The above implementations were developed using Python 3.7.3, C, and Bash scripting. These were representative edge computing methods that could extract different features (MFCC, signal energy and sample entropy) from audio signals. Other feature extraction algorithms which could be run on the constrained environment of an RPi could be easily incorporated as additional methods. The discarding of raw audio data ensured patient privacy and speaker identification was not possible. Furthermore, we did not record the speaker’s pitch information in the audio snippets or deploy methods to determine if a given window contained voiced speech.

#### 3.1.6. Human Location Tracking via Bluetooth

The Bluetooth scanning system utilized the onboard Bluetooth receiver of an RPi. In this work, we tested the *Smart Beacon SB18-3* by *kontact.io* with an RPi 4 model B. We leveraged the *scanner* package by *bluepy—a Bluetooth LE interface for Python* [83] for this purpose. The code implementation was done in Python 3.7.3. The software recorded the received signal strength indicator (RSSI) value from all the beacons transmitting the Bluetooth signal in the vicinity. We used the media access control (MAC) addresses of the Bluetooth beacons to identify them. A Python script would poll for RSSI values from all the beacons in the vicinity at regular intervals. The received RSSI values, the unique MAC address, and the recording timestamp were dumped into a file.

#### 3.1.7. Ambient Light Intensity Assessment

In this work, we used the Waveshare TCS34725 Color Sensor (Figure 1) in conjunction with an RPi to capture ambient light intensity. Among other signals, the color sensor captured the following signals which were of interest to us: (1) Red, Green, and Blue values in the RGB888 format (8-bit representation for each of the three color-channels); (2) Illuminosity in Lumen per square meter (LUX); and (3) Color Temperature in Kelvin. The RGB values and the color temperature gave us information about the ambient light color, and the illuminosity values gave us information about the ambient light intensity.

#### 3.1.8. Temperature and Humidity Detection

For temperature and humidity detection we used the DHT22 Temperature-Humidity sensor module (Figure 1) in conjunction with an RPi. The DHT22 sensor comprised a thermistor and a capacitive humidity sensor that measured the surrounding air to provide calibrated temperature and humidity values. Further, the sensor module came with a digital board that housed three pins, namely VCC, GND, and OUT. The sensor had an operating voltage of 3.3/5 V (DC), and the OUT could be read from a GPIO pin on the RPi. The temperature range was −40 to 80 °C, and the humidity range was 0–100% RH.

#### 3.1.9. Thermal Camera-Based Temperature Measurement

In [19], we showed that it is possible to perform febrile state detection using the combination of an RPi camera and the FLIR Lepton 3.5 Radiometry Long-Wave Infrared Camera with its associated Input-Output module. This system has been included in the codebase without further experimental validation.

### 3.2. Data Fusion

The following data modalities were captured at a sampling frequency of 1 Hz: (1) PIR sensor-based human movement; (2) IR camera-based human movement; (3) Audio data; (4) Bluetooth RSSI signal; (5) Ambient light intensity; and (6) Temperature and humidity. A single main script facilitated the capture of all the above data modalities and the individual time stamps corresponding to each sample. The human pose signal was recorded in an ad-hoc manner when the algorithm detected a human body. Nevertheless, the corresponding timestamps were recorded using a single clock onboard the RPi to ensure all data modalities were recorded synchronously. Further, the recorded data was easy to access via a simple directory structure consisting of separate folders for each data modality. The data collection was performed in parallel by each RPi and transmitted in real-time to a HIPAA compliant central server, which aggregated the data to perform patient state analysis.

### 3.3. Applications

The applications of our edge computing and ambient data capture system range from monitoring patient sleep in sleep labs to tracking neurodegenerative patients at their homes. In our work, we describe five experiments to demonstrate our system’s utility:Estimating occupancy and human activity phenotyping: *This utility enables us to perform patient sleep monitoring, human location tracking and activity phenotyping.*Medical equipment alarm classification using audio: *This utility facilitates patient monitor alarm monitoring in ORs or ICU rooms, where there are many system not centrally integrated.*Geolocation of humans in a built environment: *We can track humans in a built environment and model social distancing for quantifying epidemic disease exposure [84].*Ambient light logging: *This system can be used to study the effect of ambient light on human circadian rhythm.*5.Ambient temperature and humidity logging: *We can perform long-term monitoring of the effects of ambient environmental conditions on patient behavior.*

#### 3.3.1. Estimating Occupancy and Human Activity Phenotyping

In this experiment, we developed and tested algorithms to (1) Track the number of people in a given video; and (2) Differentiate people standing still from people performing a hand exercise (activity). For this, we utilized PoseNet to record keypoint locations of stick figures of humans in the video. We recorded a three-minute structured video in employing three subjects (S1, S2 & S3). The data collection was performed as illustrated in Table 1.

The video frames and the corresponding keypoints were retrospectively processed to compute the number of people in each frame. For every video frame, the corresponding keypoints file contained the x–y coordinates of the stick figures. For each human being, a separate x–y coordinate array was stored. Thus, by calculating the number of non-zero arrays in each keypoints file, we counted the number of people present in the corresponding video frame. A drawback of this approach is that it does not work well when there is occlusion. Occlusion refers to when two or more objects (in our case, humans) seemingly merge making it difficult to detect the objects in isolation [85]. A simple solution to the overcome occlusion would be to track objects temporally and perform interpolation (in time) to fill in the missing keypoints (due to occlusion). We have not included this step as all participants were stationary and far apart in our experiments (see Figure 2). However, we will address the issue of occlusion for estimating occupancy in our future work. It is important to note that the presented method allows for both human detection and localization (via keypoints). However, we do not utilize the localization information for occupancy estimation.

For human activity phenotyping, we considered the case of contrasting a standing human from an exercising human. For this, we considered the following four keypoints amongst the 17 keypoints recorded by PoseNet: (1) Left Elbow; (2) Right Elbow; (3) Left Wrist; and (4) Right wrist. In our retrospective analysis, we computed the distance (in pixels) between frames for all four keypoint locations. A distance-vector was formed for each of the four keypoint locations for all three subjects separately when they were standing and exercising. Further, we computed the interquartile range (IQR) of these distance vectors and plotted these values for all four locations for each subject both while standing and exercising. The IQR is a suitable method for suppressing outliers and capturing the spread of the data. The statistical variance, on the other hand, is affected by outliers. Hence, we chose to compute IQR over the variance to measure the spread of the data.

#### 3.3.2. Medical Equipment Alarm Classification Using Audio

In this experiment, we tested the utility of the audio feature extraction methods (energy in mel frequency bins and MFCC) proposed by us for clinical audio classification. For this, we utilized an external clinical audio database, extracted the proposed features, and performed multi-class classification.

*Dataset.* We analyzed the International Organization for Standardization/International Electrotechnical Commission (ISO/IEC) 60601-1-8 type medical equipment alarm sounds [86]. The alarm sounds comprised eight categories: general, oxygen, ventilation, cardiovascular, temperature, drug delivery, artificial perfusion, and power failure. Further, each category had two alarm sounds, namely, medium priority alarm and high priority alarm. The medium priority alarm sounds were about one second long and consisted of three musical notes that were played once, whereas the high priority alarm sounds were about 4.5 s long. They consisted of five musical notes that were played twice. All the alarm audio recordings were single channel, sampled at 22,050 Hz, and recorded in the *Waveform* audio file format. Nine musical notes were used to construct these 16 different alarm audio recordings. Table 2 lists these nine musical notes with their fundamental frequencies. The works [86,87] provide more information on the individual alarm sound recordings.

*Feature Extraction.* We used the audio data capture software described in Section 3.1.5 and computed 20 MFCC features and 10 filter bank energy features on 30 millisecond snippets of the 16 alarm sound recordings. Further, we computed STFT coefficients for the audio clips and annotated each snippet to belong to one of the following 11 classes: {Empty, $C_{4}$, $D_{4}$, $E_{4}$, $F_{4}$, $F_{4}^{\#}$, $G_{4}$, $A_{4}$, $B_{4}$, $C_{5}$, Transition}. The *Empty* class was assigned when all the STFT coefficients of a snippet were equal to zero. If a particular audio snippet was partially made up of a specific note with the rest of the samples equal to zeros, such windows were annotated as the *Transition* class. We annotated the musical notes by comparing the fundamental frequency in STFT with the values shown in Table 2. Moreover, we used the note sheets provided in [86,87] to confirm our annotations. We had a total of 1965 data points. Table 2 further provides the breakdown of the number of data points in each class.

*Classification.* Using the 30 features described, we performed an 11 class classification using five-fold cross-validation and an XGBoost [88] classifier. All codes were written in Python 3.6.3 and XGBoost was implemented using the package provided in [89]. The following hyperparameters were used without any tuning: *n_estimators* =150, *objective*=*‘multi:softmax’*, *num_class*=11, *max_depth*=6. All other hyperparameters were set to their default values. As illustrated in Table 2, the dataset contained class imbalance. We thus used both the macro averaged F1 (F1−macro) score and the micro averaged F1 (F1−micro) score as the measures for assessing classification performance. The F1−macro score gives equal importance to each class irrespective of the number of samples in each class thus providing a balanced assessment of the multi-class classification performance when the dataset is imbalanced. The F1−micro score on the other hand aggregates samples from all classes before computing the F1 score. Please refer [90] for the individual expressions for computing the two F1 scores.

*Speech Mixing.* Next, we measured the performance of the note classification algorithm in the presence of speech. For this, in addition to the ISO/IEC 60601-1-8 dataset, we used a speech record consisting of five speakers [four male and one female] from the Oxford Lip Reading Sentences 2 dataset [91]. First, we resampled the speech record to match the sampling frequency of the alarm audio recordings (22,050 Hz) and extracted the first channel of this resampled speech record, denoted by S. Next, for each of the 16 alarm audio records $A_{i},i\in \left[1,16\right]$, we uniformly randomly pick an audio snippet from the speech record $S_{i}$ which was of the same length as $A_{i}$. We then generated 10 audio records per alarm audio recording as follows:(6)$$M_{i}\left[\alpha \right]=\alpha *S_{i}+\left(1-\alpha \right)*A_{i}$$
where $M_{i}\left[\alpha \right]$ was the mixed audio recording for a given α and α∈{0,0.1,0.2,⋯0.8,0.9} was the mixing parameter that combined speech recording with alarm audio recordings. Note that * and + denoted scalar multiplication and sample-wise addition, respectively. When α was equal to 0, we had no speech component, and thus, $M_{i}\left[0\right]$ was equal to the original alarm audio recording $A_{i}$. As α increased from 0 to 0.9, the speech component in $M_{i}\left[\alpha \right]$ increased linearly, and the alarm audio component decreases linearly.

We obtained a total of 160 different audio recordings (10 mixed audio recordings per clean alarm audio recording). We re-computed 20 MFCC features and 10 filter bank energy features for these 160 audio recordings using 30 millisecond Hanning windows and a 50% overlap and obtained a total of 19,650 feature vectors. The ground truth labels for the feature vector at different α values were the same as those for α=0. Utilizing these 19,650 feature vectors and corresponding labels, we performed an 11 class classification of musical notes using five-fold cross-validation and XGBoost [88] classifier. The hyperparameters were the same as it was when there was no speech mixing.

#### 3.3.3. Geolocation of Humans in a Built Environment

In this experiment, we processed the RSSI signal received by the RPi to perform room-level location detection of humans using a Bluetooth beacon. We set up nine RPis in a built environment where each RPi was loaded with the software to capture RSSI values as received from a specific Bluetooth beacon. The built environment consisted of three rooms, and three RPis were present in each room, approximately equidistant from the center of the room. A *Kontakt.io* Bluetooth beacon with a unique MAC address was used in the experiment. A human subject carried the Bluetooth beacon in their clothes and moved around the space as illustrated in Table 3. Please note that this is a minor limitation of our system and the usage of a Bluetooth beacon (albeit inside one’s clothes) resulted in our system not being completely non-contact.

The processing of the collected RSSI values to perform geolocation of humans was done on a central server. This was because we collected data from multiple RPis to perform geolocation. Once the data was transferred to the cloud from each RPi, we downloaded the data onto a central server and performed geolocation. The RSSI signals captured by each RPi were non-uniformly sampled. Hence, these signals were converted to a uniformly sampled signal with a sampling frequency equal to 1 Hz by filling missing data using the following equation.
(7)$$\begin{array}{c} \begin{array}{c} RSSI_{n}\left[currentTime\right]=RSSI_{n}\left[previous\right]\times max\left(\left[\beta \times \left(currentTime-previous\right),1\right]\right) \end{array} \end{array}$$
where, n∈[1,9] was the index variable to recognize RPis, $RSSI_{n}$ was the vector of RSSI values captured by $n^{th}$ RPi, *currentTime* was the time (in seconds) at which we did not have a reading of the RSSI value, *previous* was the closest predecessor time point (in seconds) to *currentTime* at which we had a reading of the RSSI value, β was the decay parameter that controlled the rate at which the RSSI value decayed when RSSI values were missing and *max* was a function to compute the maximum value in the input vector. The time difference (*currentTime-previous*) was expressed in seconds.

Further, any RSSI value less than −200 dBm was clamped to −200 dBm to have all RSSI values in a fixed range. We set β to 0.2, which corresponded to maintaining the previous value for 5 s before the RSSI values were decayed when the RSSI values were missing. Further, we computed the average RSSI signal for each room by computing the mean value of the RSSI signals captured by the three RPis in each room. We used the softmax function to obtain a probability vector that gave the probability of the human subject with the Bluetooth beacon to be present in each of the three rooms at any given point in time. The averaging of RSSI values from multiple RPi receivers and the further usage of the softmax function significantly suppressed the effect of noisy RSSI samples.

#### 3.3.4. Ambient Light Logging

To perform ambient data logging, we set up an RPi with the Waveshare TCS34725 color sensor in a built environment. The RPi was loaded with the associated software needed to capture the ambient light intensity values. Table 4 provides a timeline of events that occurred during ambient light data capture. The duration between 1 a.m. and 2 a.m. was reserved for data upload, and no data capture was performed during this period. The color sensor was set up in a place that received natural sunlight during the day and received light from light sources in the room during the night. The lights in the room were turned ON when the natural sunlight was not adequate for a normal human lifestyle. The lights in the room remained ON until “sleep time” of the residents in the built environment when the lights were turned OFF. We recorded the ambient light data for two consecutive days. In parallel, we tracked the weather conditions of the data collection site and recorded the minute-to-minute local cloud cover information. With this setup, we studied the effect of cloud cover, sunrise and sunset times, artificial lights in the room, and buildings around the data collection site on the ambient light data captured by the color sensor. We divided the entire time period into 4 sections: $D_{clear}$, $N_{1}$, $D_{cloudy}$, $N_{2}$. Here, $D_{clear}$ represents the day-period (sunrise to sunset) on the first day when the skies were clear, $N_{1}$ represents the first night (sunset to sunrise), $D_{cloudy}$ represents the day-period (sunrise to sunset) on the second day when the skies were extremely cloudy (average cloud cover > 80%), and $N_{2}$ represents the second night (sunset to sunrise). Since the cloud cover information during the night had little or no effect on the ambient light intensity in the room, in our analysis, we only used the cloud cover data tracked during $D_{clear}$ and $D_{cloudy}$.

#### 3.3.5. Ambient Temperature and Humidity Logging

We validated the DHT22 temperature and humidity sensor against a commercially available sensor in this experiment. We set up an RPi with the DHT22 sensor in a built environment. The RPi was loaded with the necessary software to continuously capture temperature and humidity values and the associated UTC timestamps. We collected the temperature ($T_{RPi}$) and humidity ($H_{RPi}$) values with this setup at a sampling frequency of 1Hz for three consecutive days (about 72 h). Further, we set up the *ORIA mini Bluetooth Temperature-Humidity sensor* (a commercial sensor) beside our RPi setup and simultaneously performed temperature ($T_{cs}$) and humidity ($H_{cs}$) measurements using the commercial sensor. The commercial sensor allowed the export of the recorded $T_{cs}$ and $H_{cs}$ measurements in the form of comma-separated value files via an Android application. The commercial sensor output contained measurement values at a sampling rate of 0.001667 Hz (one sample per 10 min). Hence, we retrospectively processed the $T_{RPi}$ and $H_{RPi}$ measurements captured by the DHT22 sensor to match the number of samples and the measurement timestamps corresponding to the commercial sensor via the following procedure. For every timestamp ($ts_{cs}$) at which we had the temperature and humidity values from the commercial sensor, we constructed a 10-min window which spanned from $ts_{cs}-(10$ min) to $ts_{cs}$. We collated all $T_{RPi}$ and $H_{RPi}$ measurements in this time window and computed the mean value of these measurements to obtain $T_{RPi-\mu }\left[ts_{cs}\right]$ and $H_{RPi-\mu }\left[ts_{cs}\right]$. We then compared the $T_{RPi-\mu }$ with $T_{cs}$ and $H_{RPi-\mu }$ with $H_{cs}$ by plotting the signals one over the other. Further, we performed correlation analysis and fit separate linear models for the temperature and humidity measurements. Finally, we created separate Bland-Altman plots for the temperature and humidity measurements.

## 4. Results

### 4.1. Estimating Occupancy and Human Activity Phenotyping

Figure 3a shows the comparison of the algorithm’s estimation for human occupancy in each video frame with respect to the ground truth values. We obtained an accuracy of 94% for the occupancy estimation experiment. Our performance on the occupancy estimation task was comparable to exisiting works as shown in Table 5. For contrasting humans performing hand exercises from humans standing still, we visualized the IQR values of the frame-to-frame distances (in pixels) for four keypoints corresponding to the human hand. The IQR values were always larger for exercising than standing still for all four keypoints and all three subjects. When a subject was standing still, the keypoints barely moved and thus would typically have low values (<5 pixels). The individual IQR values for all three subjects at the four keypoint locations are illustrated in Figure 3b. The IQR-based features were linearly separable for the two classes (standing vs. exercise) which translates to an accuracy of 100%. However, with a larger dataset we expect the classification performance to reduce. Our performance on the activity recognition task was comparable to existing works as illustrated in Table 5.

### 4.2. Medical Equipment Alarm Classification Using Audio

Table 6 shows the results for the 11-class medical equipment alarm note classification. We computed the micro averaged (F1−micro) and the macro-averaged (F1−macro) F1 scores for the two experiments. When there was no speech content in the alarm audio recordings, we obtained an F1−micro equal to 0.98 and an F1−macro equal to 0.97. Retraining with speech resulted in a drop of 5.1% and 6.2% in F1−micro and F1−macro scores, respectively.

### 4.3. Geolocation of Humans in a Built Environment

The geolocation computation using the RSSI data was performed on a central server. Figure 4a shows interpolated RSSI values captured by each of the nine RPis that were placed in the built environment, and Figure 4b illustrates the corresponding probability of the subject being in rooms 1, 2, or 3. The ground truth of the subject’s presence is shown using translucent colors in the background. Specifically, translucent blue denoted being present in room 1, translucent green denoted being present in room 2, and translucent red denoted being present in room 3. Further, the transition from one room to another was illustrated by overlapping colors corresponding to the two rooms. It is evident from Figure 4 that our system did an excellent job of identifying the room in which the person was present. Specifically, for 592 out of 600 s, the human tracking system correctly identified the subject’s presence in one of the three rooms, which corresponded to an accuracy of 98.67%. We compared the performance of the geolocation task with other existing works (see Table 5). The performance of our technique was comparable to exisiting work in the field.

### 4.4. Ambient Light Logging

Figure 5 depicts the variation of ambient light intensity over two days. The minimum, median, and maximum illuminance values during the $D_{clear}$ period were equal to 0.56 LUX, 71.08 LUX, and 186.92 LUX, respectively, whereas the minimum, median, and maximum illuminance values during the $D_{cloudy}$ period were equal to 0.26 LUX, 26.93 LUX, and 117.41 LUX, respectively. Thus, the median difference in illuminance between the clear and cloudy days was equal to 44.15 LUX. The minimum, median and maximum illuminance between the lights-ON and lights-OFF times was equal to 15.08 LUX, 15.30 LUX, 16.66 LUX on the first night ($N_{1}$) and equal to 9.39 LUX, 10.15 LUX, 15.02 LUX on the second night ($N_{2}$). The illuminance was consistently equal to zero between the lights-OFF and sunrise times. Further, we observed a dip in illuminance when the Sun hid behind a skyscraper and cast a shadow on the data collection site during the day.

### 4.5. Ambient Temperature and Humidity Logging

Figure 6 illustrates the comparison between the processed temperature and humidity values from the DHT22 sensor with the outputs from a commercial sensor. The temperature values from the two sensors closely followed each other with a root mean squared error (RMSE) between the two measurements equal to 0.28 ${}^{{}^{\circ}}$C and a coefficient of determination (r${}^{2}$) equal to 0.97. Over 97% of the samples lay within the limits of agreement [−1.96 SD, +1.96 SD] in the Bland-Altman plot. Further, the mean difference was equal to −0.4 ${}^{{}^{\circ}}$C. The humidity values from the two sensors closely followed each other with an RMSE between the two measurements equal to 1.00%RH and an r${}^{2}$ equal to 0.90. Over 95% of the samples lay within the limits of agreement [−1.96 SD, +1.96 SD] in the Bland-Altman plot. Further, the mean difference was equal to −1.2%RH.

## 5. Discussion

The work described in this article aims to extend the types of data found in traditional clinical monitoring environments and provide a simple system to capture data in the built environment, outside of clinical settings. Many commercial (clinical and consumer) systems are either designed to keep data in a proprietary ‘walled-garden’ to reduce competition or are not designed for the high throughput needed to transmit/record the data. The RPi-based edge computing system described in this work allows direct data import via USB and upload to the cloud asynchronously to overcome these issues.

We have included methods to capture audio data, physical movement, and location of subjects. As we have demonstrated, audio data allows capturing of all alarms in the clinical space. While some monitors transmit some of these events or signals over the network, it is often costly or impossible to gain access to such data, and data integration and synchronization are highly problematic. Moreover, such systems do not provide a holistic picture of the environment. For example, the volume of the alarm, together with the background noise, contributes to noise pollution and has been shown to affect caregivers and patients alike [4,5,92]. Beyond alarms in the clinical environment, it is possible to capture whether a patient is being mechanically ventilated (and at what frequency), groans and expression of pain, and other non-verbal utterances. It is possible to add speaker and voice recognition to the code base, to identify who is speaking and about what, providing insight into clinical (and non-clinical) discussions that may provide additional diagnostic power. For instance, by differentiating patients from family members, it is possible to assess both the level of clinical team support and frequency of bedside visits and the social support that a patient may have (inferred by the number and duration of visitations by friends and family). Tracking the time clinicians spend with patients and the level of expertise available could help identify gaps in care. In addition, by tracking Bluetooth transmitter strength of body-worn devices (e.g., badges or phones), it is possible to infer motion, an individual’s identity (through a look-up table), and even the exact location of an individual if more than one Bluetooth receiver base is used. Real-time and accurate tracking of humans using Bluetooth beacons needs a receiver (RPi) sensor network. Further, we can have a central server where all the RPis communicate and update the collected RSSI values. We can then have algorithms operating on this database in real-time to perform the geolocation of humans. We have implemented this system in a clinical environment at Emory Healthcare, Atlanta, USA, to monitor the real-time location of humans.

By capturing motion via video, we can probe even deeper into assessing the patient and their environment. For instance, we can estimate the quantity of sleep, sedation, and agitation that a patient experiences, all of which have been linked to recovery [93]. At the same time, if the motion is associated with clinical care, then the intensity of activity can indicate when treatments, observations, or specific activities (such as replacing drips) took place. While we know that the ratio of nursing staff to patient impacts outcomes [7], there are no studies that examine the time at the bedside and the actions taken at the bedside in terms of their impact on the outcome. However, it is known that time spent at the bedside is linked to improved patient satisfaction [94].

Finally, the data modalities we capture provide us a unique opportunity to perform multimodal analysis of the patient state. For instance, consider the case of human sleep monitoring in a home environment. All we need to do is to place the Bluetooth beacon in the patient’s clothes before they sleep. The motion signals captured during the patient’s sleep give us the timestamps when the patient moved in the bed. Based on the intensity of the motion signal, we can delineate minor movements (rolling over) from major movements (sitting up in bed). The simultaneous recordings of audio-features and illuminance, which can act both as sleep inhibitors (flushing toilet or turning ON lights) and wake-event markers, give us valuable information about the patient’s sleeping patterns. Further, the RSSI signal analysis will provide us with all the times that the patient exits the room during the night. Finally, via a long-term monitoring protocol, we can recommend ideal sleeping conditions to the patient by monitoring the temperature and humidity in the room. Thus, the system presented here provides a low-cost method for performing deep analysis, both at home and in a clinical setting. The system itself has been deployed for patient monitoring and data collection in two separate healthcare facilities located in the United States of America: (1) A New York state department of health funded center for excellence facility that offers residential, medical, clinical and special education programs to the residents (25 units); and (2) A Mild Cognitive Impairment rehabilitation program facility at Emory Healthcare in Atlanta (40 units). For a 12 h (7 a.m. to 7 p.m.) recording of all data modalities (except PoseNet), a total of 832 MB of data was recorded. In both these facilities, data collection is ongoing. We are capturing simultaneous human movement and pose signals, privacy preserved audio features and ambient environmental signals using the system described in the current work (see Section 3). The future work involves performing multimodal analysis and the applications described in our work on the collected (larger) dataset. Further, we will discover features that help detect patients with high-severity neurological diseases.

## 6. Conclusions

This article presents an overview of a generalized open-source system for edge computing in clinical and home environments. It provides real-time data elements and analysis that are not generally present in electronic medical records yet are associated with clinical performance, diagnosis, and outcomes. In particular, we focused on the acoustic environment (such as speech, alarms, and environmental noise), human movement detection and geolocation (absolute or relative to others). We also added camera-based analysis for occupancy estimation and environmental sensors (temperature, humidity, and light). Further, we included methods for privacy-preserving feature extraction to provide a generally acceptable system that is unlikely to violate hospital policies and other privacy regulations, which may reduce the anxiety of administrators and clinicians concerning the level of monitoring. Encryption and data transfer protocols were not included as these are specific to each institution. The implementation on a state-of-the-art extensible edge computing system at a relatively low cost provides a high degree of flexibility in the design. The bill-of-materials and open-source code to replicate the work described here have been made publicly available under an open-source license [73].

## Figures and Tables

**Figure 1 sensors-22-02511-f001:**
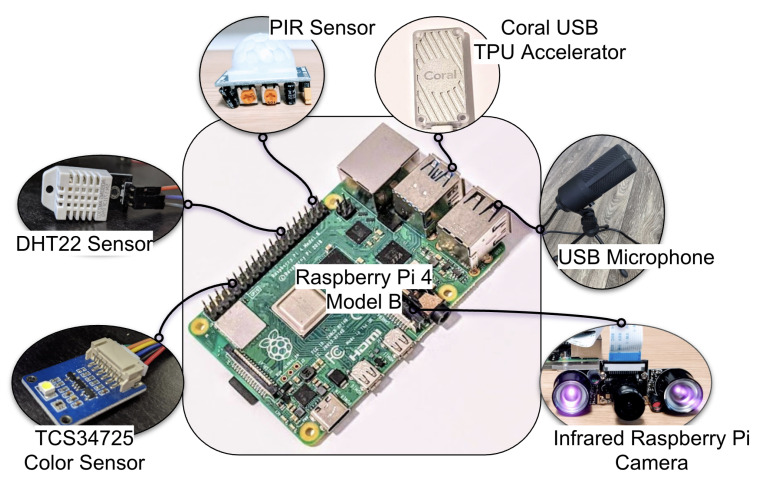
Edge computing and ambient data capture system. (PIR: Passive Infrared Sensor; USB: Universal Serial Bus; TPU: Tensor Processing Unit).

**Figure 2 sensors-22-02511-f002:**
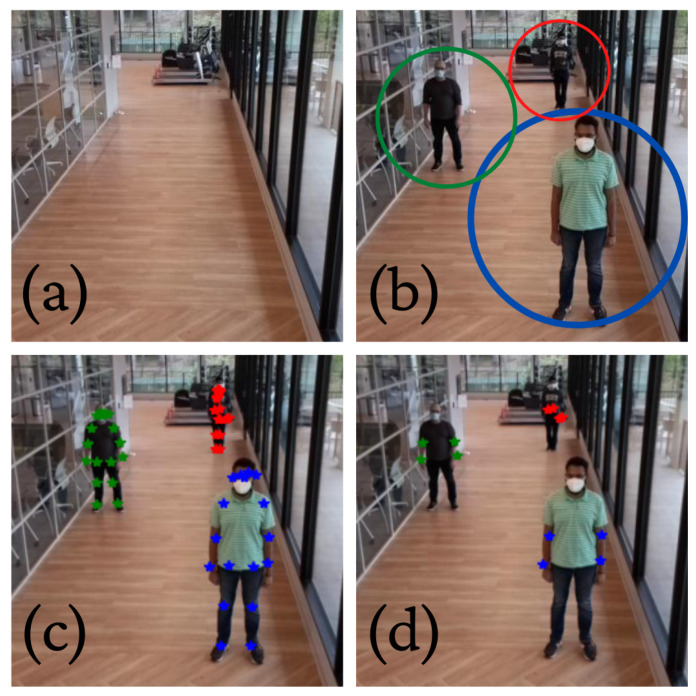
PoseNet [75] for human pose detection. (**a**) We show the scene without any humans. (**b**) The scene with three humans standing. (**c**) The identified keypoints using PoseNet have been overlaid on the individual humans. (**d**) The keypoints corresponding to the elbows and wrists have been retained. This is useful for analyzing hand movements.

**Figure 3 sensors-22-02511-f003:**
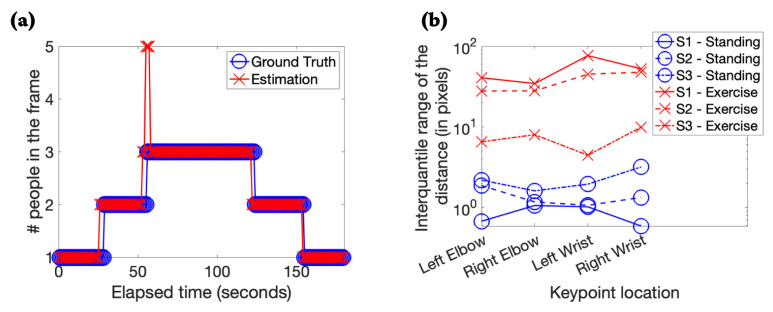
Results for occupancy estimation and activity phenotyping. (**a**) The comparison between the ground truth and the estimation by the algorithm for the number of people in each video frame is shown. (**b**) We show the interquartile range of the distance (in pixels) between the keypoints from consecutive frames (sampling frequency = 1 Hz) for three subjects standing and performing exercise.

**Figure 4 sensors-22-02511-f004:**
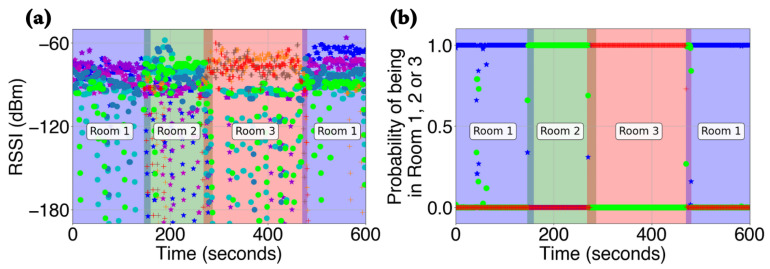
Human geolocation via Bluetooth. The translucent background colors indicate the ground truth values, and the corresponding room number has been labeled. (**a**) The received signal strength indicator (RSSI) values as captured by each of the nine Raspberry Pis during the experiment are shown. It is measured in decibels with reference to one milliwatt (dBm). (**b**) The corresponding probability values of the subject being in Rooms 1, 2, or 3 during the experiment. The blue color depicts Room 1, the green color depicts Room 2, and the red color depicts Room 3.

**Figure 5 sensors-22-02511-f005:**
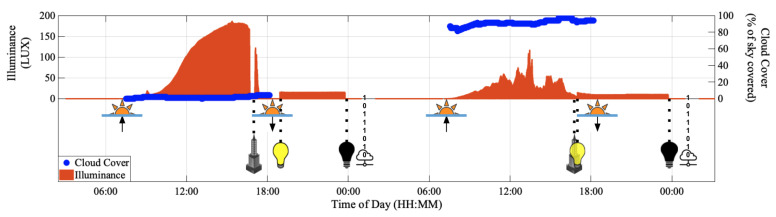
Observing the effects of cloud cover, sunrise, sunset, lights-ON, lights-OFF and buildings around the data collection site on the ambient light in a built environment for two continuous days (48 h). The solid-orange bars (❚) depict the amount of ambient light sensed by the color sensor in Lumen per square meter (LUX). The solid-blue circles (●) depict the local cloud cover in percentage of local sky covered by clouds. The sunrise and sunset times are indicated with the rising and setting sun symbols, respectively, using the upward and downward arrows. The yellow and black bulbs specify the lights-ON and lights-OFF times, respectively. The skyscraper symbol indicates the time when the Sun goes behind a skyscraper and causes a shadow onto the location where ambient light was being tracked. Data upload is depicted by binary values and a cloud node.

**Figure 6 sensors-22-02511-f006:**
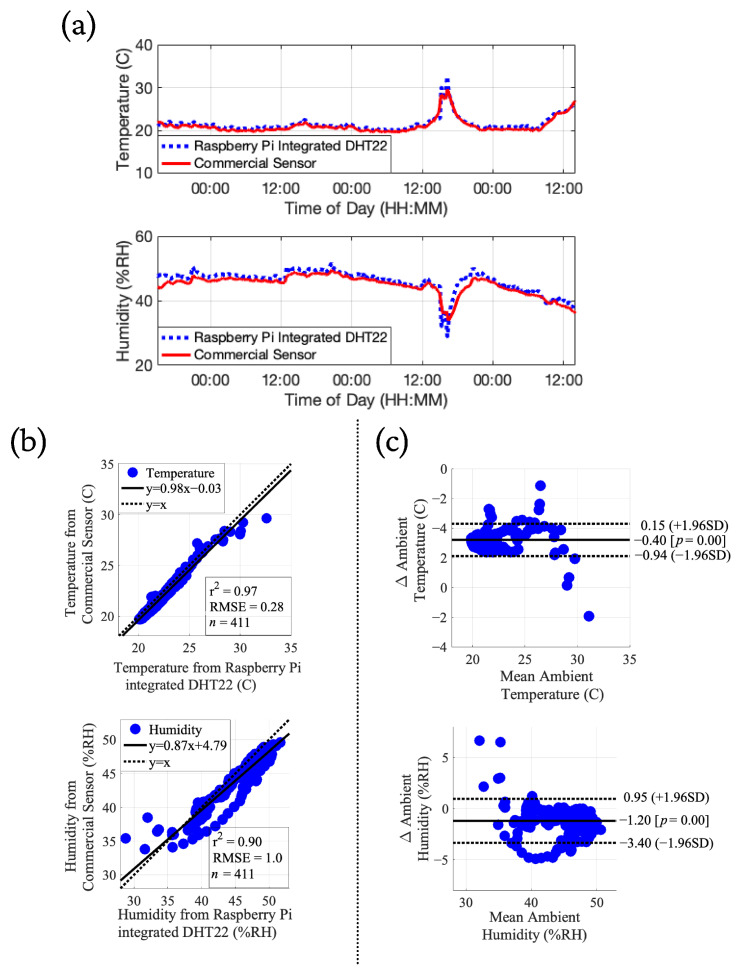
Comparison of temperature and humidity values captured by the Raspberry Pi (RPi) integrated DHT22 sensor and a commercial sensor. The top plot corresponds to temperature in each subplot, and the bottom plot corresponds to humidity values. (**a**) The dashed blue lines indicate the processed temperature and humidity values captured by the RPi integrated DHT22 sensor. The solid red lines indicate the corresponding values captured by the commercial sensor. (**b**) The correlation plots between values captured from the DHT22 sensor and the commercial sensor. The solid-blue circles (●) indicate individual temperature and humidity tuples. The linear fits on the data and their deviations from the 45${}^{\circ }$ line are depicted in the plot. (**c**) The Bland-Altman plot between the two measurements. The solid black lines indicate the mean difference between the two measurements. The dashed black lines indicate the +1.96 and the −1.96 standard deviation (SD) lines for the difference between the two measurements.

**Table 1 sensors-22-02511-t001:** Estimating occupancy and activity phenotyping.

Start Time	End Time	Duration	Action
(s)	(s)	(s)	
0	30	30	One person standing
30	60	30	Two people standing
60	90	30	Three people standing
90	120	30	Three people exercising
120	150	30	Two people exercising
150	180	30	One person exercising

**Table 2 sensors-22-02511-t002:** Class labels including the Musical Notes in International Organization for Standardization/International Electrotechnical Commission (ISO/IEC) 60601-1-8 Alarm and their count [*E* stands for the *Empty* class and *T* stands for the *Transition* class].

Musical Note	Fundamental Frequency (Hz)	Count
$C_{4}$	261.63	207
$D_{4}$	293.66	43
$E_{4}$	329.63	32
$F_{4}$	349.23	46
$F_{4}^{\#}$	369.99	27
$G_{4}$	392.00	66
$A_{4}$	440.00	45
$B_{4}$	493.88	17
$C_{5}$	523.25	81
*E*	-	969
*T*	-	432

**Table 3 sensors-22-02511-t003:** The human tracking experiment.

Start Time	End Time	Duration	Action
(s)	(s)	(s)	
0	146	146	Stay in room 1
146	159	13	Move from room 1 to room 2
159	268	109	Stay in room 2
268	286	18	Move from room 2 to room 3
286	470	184	Stay in room 3
470	480	10	Move from room 3 to room 1
480	600	120	Stay in room 1

**Table 4 sensors-22-02511-t004:** The ambient light tracking experiment.

Day	Time (HH:MM)	Action
Clear Day ($D_{clear}$)	07:32	Sunrise
	18:10	Sunset
Night-1 ($N_{1}$)	18:55	Lights-ON
	23:45	Lights-OFF
	01:00	Data upload start
	02:00	Restart data collection
Cloudy Day ($D_{cloudy}$)	07:31	Sunrise
	16:58	Lights-ON
	18:11	Sunset
Night-2 ($N_{2}$)	23:42	Lights-OFF

**Table 5 sensors-22-02511-t005:** A comparison of existing systems for non-contact human monitoring with our methods [KNN: K-Nearest Neighbour; RGB: Red-Green-Blue; RNN: Recurrent Neural Networks; DNN: Deep Neural Network; CNN: Convolutional Neural Networks; NA: Not Available; BLE: Bluetooth Low Energy; RO-EKF-SLAM: Range-Only Extended Kalman Filter Simultaneous Localization and Mapping; IMU: Inertial Measurement Unit].

Task	Reference	Sensor	Method	# People	Accuracy
Human occupancy estimation	Tyndall et al. [61]	Thermal Imager	KNN	3	82.6%
	Ahmad et al. [63]	RGB Camera	RNN	4	93.4%
	Metwaly et al. [62]	Thermal Imager	DNN	5	98.9%
	Our work	RGB Camera	RPi-PoseNet	3	94.0%
Human activity recognition	Zerrouki et al. [64]	RGB Camera	AdaBoost	17	94.9% (6-Activities) ^1^
	Singh et al. [67]	Thermal Imager	Random Forest	3	97.4% (4-Activities)
	Zhao et al. [65]	RGB Camera	CNN	NA	90.4% (3-Activities) ^2^
	Park et al. [66]	RGB Camera	DNN	6	88.9% (4-Activities)
	Our work	RGB Camera	RPi-PoseNet	3	100% (2-Activities)
Geolocation of humans in a built environment	Sato et al. [68]	BLE	RO-EKF-SLAM	6	Mean Error =2.73 m
	Martín et al. [69]	BLE+IMU	KNN	4	96.4% (6-Rooms)
	Our work	BLE	Softmax	1	98.7% (3-Rooms)

^1^ We have averaged their results on the two datasets provided in the paper. ^2^ We have excluded the “Empty” class and do not consider it as human activity.

**Table 6 sensors-22-02511-t006:** Note Classification in Medical Equipment Alarm.

Setting	F 1−micro	F 1−macro
Without Speech	0.98	0.97
With Speech Mixing	0.93	0.91

## Data Availability

Not applicable.

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
