# Peer review of "An Edge Computing and Ambient Data Capture System for Clinical and Home Environments"

_sensors, 2022, doi:10.3390/s22072511_

Round 1
Reviewer 1 Report
The following aspects must be improved:
- section with state of the art must contain descriptions of other similar systems, too
- the novelty of the proposed system must be provided more clearly in comparison with other existing ones
- obtained results must be compared with other existing ones
- tracking of persons must be explained more clearly; what should it happen if a person is occluded by another ones
- the same question in case of occupancy calculation
- how were identified human activities: exercise, standing?
- figures and tables must be placed before their first reference in the text.
Reviewer 2 Report
This paper presents a low-cost sensor platform based on a raspberry pi for monitoring clinical and home environments. Several sensors (passive infrared sensor, infrared camera, microphone, temperature-humidity, and luminosity) are described and used in five different application. The paper is mostly understandable and well-written, but there is some flaws that should be fixed in order to improve its content and relevance for a wide audience.
1. Sensors are widely described as well as the way in which data is captured and processed in each case. Nevertheless, it is not clear enough if the application of these sensors is innovative or they have been already used in other works. The section Related work focuses on applications but not in the use of these sensores for those applications. Due to the low cost of these sensors, it is expected that they have been used in other works (that should be mentioned here).
2. If the contribution of this work is the whole platform and the five sensors working together, then some applications and experiments should be provided where all the sensors work effectively and simultaneously. The applications included in the paper use only one (or two) sensor, thus the potential contribution of the platform is not clear.
3. In section 3.1.2, authors claim "we used the captured movement data to build a binary classifier for obstructive [...] equal to 91%". This development and validation are not described in this work. If it belongs to another work, please include a reference.
4. Bluetooth beacons carried by people is not aligned with the proposal of non-contact monitoring (e.g., patients sleeping with a Bluetooth beacon). Consider modify the rationale of this sensor and application because it does not follow the requirements (unobstrusiveness and non-contact) of the system. Moreover, using 9 RPis in a built environment to collect RSSI from one beacon does not seem scalable or low-cost.
5. The contribution of sensors of humidity and temperature does not seem valuable. Consider stressing their relevance or innovation.
6. If IR camera overpasses PIR sensor, why keeping the PIR sensor is not clear.
7. Cameras do not recognize people, only the number of humans in a room (ocuppancy). Thus, it cannot be argued to be used for monitoring motion of clinicians, patients or relatives.
8. In section 2.2 some paragraphs are not related work but methodology
9. In section 3.3.3, geolocation is performed in a central server, not edge computing. Results from this experiment are not included.
10 .Figure 4 is hard to understand. Consider making it clearer.
11. Authors indicate that part of the system has been deployed in real settings, but no data about these real experiences are provided.
12. Update bibliography
13. There is no Appendix A available
Reviewer 3 Report
Undoubtedly, gathering information about the patient and the conditions around them is necessary in the process of diagnosis, treatment and convalescence. More information and biomedical signals make it possible to identify the progress of recovery, to optimize the treatment strategy adopted, but in special cases to detect undesirable effects of the applied medical procedures, such as insomnia, neurosis, heart tachycardia, problems with concentration, weakness, and depression. Among the registered biomedical signals mentioned in the article and other additional data (location, movement, parameters of the internal environment), the measurement of eye movement is also a very important signal. Observation of the eye movement allows to determine the psychophysical state of the patient, his ability to concentrate, correct reaction to external stimuli (changes in lighting, the appearance of new elements in his environment), the ability to observe the space in the field of view. The measurement system proposed by the authors is an interesting technical solution to the problem of data acquisition and analysis, masking of sensitive information and data collection.
I find the article interesting, carefully prepared, and containing many necessary technical details. Introduction, the review of existing solutions is sufficiently wide to indicate the justification for the efforts made by the authors to develop and test the device. The content of the article is quite extensive in relation to the amount of new information, there are repetitions. However, I assume that the purpose of these repetitions is to make the content of the article easier to understand.
Regarding general remarks. I have comments about the device description, I miss information about the device's power source. The Raspberry Pi platform in the default configuration requires a relatively large amount of energy. Putting the system in the ultra-economy mode is not easy. I think it would be interesting to have a short discussion on the need itself and the possible ways to optimize the energy consumption of the system. Unless the authors have decided that the system will not be battery powered. So I miss the information about the criteria and factors influencing the choice of the system power supply.
I have a few comments to the content of the presented article. The order of the comments does not reflect their significance. It results only from the order of appearance in the text of the manuscript:
- Lines 88-89, “Human bodily movement causes motion artifacts in the physiological signals (say ECG).” - It's not very clear what the authors mean by “motion artifacts”.
- Line 231, “different features” - it would be nice to indicate what features.
- Line 252, “(LUX)” - 1 lx equals 1 lm / m2, not 1 lm / ft2.
- Line 303, “in the video” – the authors specified that the camera gives an image with a resolution of 8 megapixels, but what was the division into regions (mentioned in line 175).
- Line 542, “such as replacing drips” - how such an activity will be identified; How to distinguish between replacing the drip container and checking the fluid level in the container.
Round 2
Reviewer 1 Report
Since all my comments were addressed, I recommend to publish the paper.
Reviewer 2 Report
The authors have performed a remarkable effort for satisfying all the reviewer's comments. Consequently, the manuscript has been strengthened and its potential interest increased. No further comments needed.